# Low-resource Bilingual Dialect Lexicon Induction
# with Large Language Models

**Ekaterina Artemova**   **Barbara Plank**
MaiNLP, Center for Information and Language Processing (CIS), LMU Munich, Germany
Ekaterina.Artemova@lmu.de, B.Plank@lmu.de

## Abstract

Bilingual word lexicons are crucial tools for multilingual natural language understanding and machine translation tasks, as they facilitate the mapping of words in one language to their synonyms in another language. To achieve this, numerous papers have explored bilingual lexicon induction (BLI) in high-resource scenarios, using a typical pipeline consisting of two unsupervised steps: bitext mining and word alignment, both of which rely on pre-trained large language models (LLMs).

In this paper, we present an analysis of the BLI pipeline for German and two of its dialects, Bavarian and Alemannic. This setup poses several unique challenges, including the scarcity of resources, the relatedness of the languages, and the lack of standardization in the orthography of dialects. To evaluate the BLI outputs, we analyze them with respect to word frequency and pairwise edit distance. Additionally, we release two evaluation datasets comprising 1,500 bilingual sentence pairs and 1,000 bilingual word pairs. They were manually judged for their semantic similarity for each Bavarian-German and Alemannic-German language pair.

## 1 Introduction

Learning in low-resource settings is one of the key research directions for modern natural language processing (NLP; Hedderich et al., 2021). The omnipresent pre-trained language models support high-resource languages by using increasingly large amounts of raw and labeled data. However, data scarcity hinders the training and evaluation of NLP models for less-resourced languages. At the same time, the participation of native speakers of different languages in the world of digital technologies increases the demand for supporting more language varieties. This encourages studies to explore suitable transfer learning and cross-lingual techniques.

Local varieties (dubbed as dialects) may fall under the umbrella of low-resource languages. Processing dialects faces unique challenges that should be addressed from a new perspective. Large volumes of writing in dialects such as newspapers or fiction are rarely produced and access to conversational data in social media is limited and difficult to reliably collect. Besides, dialects are non-standardized, they lack unified spelling rules and exhibit a high degree of variation (Millour and Fort, 2019). Finally, dialects may additionally show a significant rate of code-mixing to standard languages (Muysken et al., 2000).

The mainstream of cross-lingual transfer research towards low-resource languages, e.g., (Muller et al., 2021; Riabi et al., 2021), builds upon cross-lingual representations, namely static embeddings (Lample et al., 2018) or multilingual pre-trained language models (Devlin et al., 2019; Conneau et al., 2020). As shown by Muller et al. (2021) various factors can influence the performance, including the degree of relatedness to a pre-training language and the script. As there is no winning technique for all languages, it is important to understand how cross-lingual representations act for each particular language or a language family and whether the results in processing standard languages are transferable to its dialects.

In this paper, we focus on the ability of cross-lingual models to make semantic similarity judgments in German and two of its dialects, namely Bavarian (ISO 639-3:bar) spoken in South Germany, Austria, South Tyrol, and Alemannic (ISO 639-3:gsw) spoken in Switzerland, Swabia, parts of Tyrol, Liechtenstein, Alsace, and Italian re-

gions. Using the available raw data in Wikipedia (Section 3) we induce two bilingual lexicons, mapping words from Bavarian / Alemannic to German. To do so, we first mine bitext sentences (Section 4) and exploit machine translation aligners next (Section 5). The output lexicon exhibits an evident tendency for a German word to be aligned to multiple dialect synonyms due to spelling variations. Finally, we manually evaluate the output of each step: we evaluate the semantic similarity in (i) 1,500 bilingual sentence pairs according to the Likert scale and (ii) 1,000 bilingual word pairs according to a binary scale. Our results demonstrate the discrepancy between natural writing and linguistic dictionaries.

To sum up, this paper explores the following **research question** (RQ): How effective are standard pipelines for inducing bilingual lexicons for German dialects, and what factors influence their performance? To answer this question we make the following **contributions:** (i) We conduct a thorough analysis of cross-lingual models' behavior in two tasks of bitext mining and word alignment for the German language and two of its dialects. (ii) We release the evaluation datasets for bitext mining (1,500 samples each) and for bilingual lexicon induction (1,000 samples which). We make mined bitext dataset and induced bilingual lexicons for the Bavarian and Alemannic dialects publicly available. (iii) We publish the code to reproduce bitext extraction and word alignment in open access.[1]

## 2   Related work

**NLP for German dialects.**   Previous efforts in processing German dialects mainly concentrate on speech processing. BAStat comprises the recordings of spoken conversational speech from main areas of spoken German (Schiel, 2010). Dogan-Schönberger et al. (2021) build a parallel corpus of spoken Alemannic dialect, in which a sentence in German is matched with spoken and written translations into eight dialects. ArchiMob is a general domain spoken corpus equipped with transcriptions and part-of-speech labeling (Scherrer et al., 2019). In the domain of written text processing, machine translation techniques have been applied to re-write sentences from dialect to standard German (Honnet et al., 2018; Plüss et al., 2020; Lambrecht et al., 2022). Other works

---

[1] https://github.com/mainlp/dialect-BLI

tackle sentiment classification (Grubenmann et al., 2018), part-of-speech tagging (Hollenstein and Aepli, 2014) and dialect identification tasks (von Däniken et al., 2020). Burghardt et al. (2016) have collected a bilingual Bavarian-German lexicon using the knowledge of Facebook users, while Schmidt et al. (2020) hire expert native speakers to build a bilingual Alemannic-German lexicon. Language resources used to collect raw dialect data are Wikipedia, social media (Grubenmann et al., 2018), regional newspapers, and fiction (Hollenstein and Aepli, 2014). For a more comprehensive review, we refer to the concurrent survey of Blaschke et al. (2023).

**Bitext mining.**   Sentence representations are core to mining bitext (dubbed as *parallel* or *comparable* datasets) in an unsupervised fashion (Hangya et al., 2018). Pires et al. (2019) show that `[CLS]`-pooling with multi-lingual encoders performs reasonably well for the task. Most recent studies proposed learning sentence embeddings from encoder-decoder models with a machine translation objective (Artetxe and Schwenk, 2019), by extending a monolingual sentence model to cross-lingual encoding with knowledge distillation (Reimers and Gurevych, 2020), or from dual encoders with a translation ranking loss (Feng et al., 2022). Bitext datasets collected from Wikipedia (Schwenk et al., 2021a) and the Common Crawl corpus (Schwenk et al., 2021b) serve to train machine translation models (Briakou et al., 2022) and to improve cross-lingual methods for structured prediction (El-Kishky et al., 2021). Chimoto and Bassett (2022) show that cross-lingual sentence models scale across unseen languages. We adopt the recent state-of-the-approach of (Reimers and Gurevych, 2020), which scores sentences embeddings, obtained from cross-lingual embedders, with cosine similarity measure in order to retrieve most similar sentence pairs.

**Bilingual lexicon induction (BLI).**   Works in bilingual lexicon induction can be cast into two groups. *Mapping-based approaches* project monolingual word embeddings into a shared cross-lingual space with a varying degree of supervision (Lample et al., 2018; Artetxe et al., 2018; Joulin et al., 2018). *Corpora-based* methods combine bitext mining with word alignment (Shi et al., 2021). Intrinsic evaluation compares

| Language | # dialects | # pages | # sent. | # tokens | #types | Manual labelling # bitext | # synonyms | Bilingual lexicon induction # bitext | # synonyms |
|---|---|---|---|---|---|---|---|---|---|
| Bavarian | 9 | 43k | 230k | 3.7mln | 350k | 1,254/1,500 | 860/1,000 | 17k | 11k |
| Alemannic | 32 | 71k | 500k | 9.5mln | 600k | 644/1,500 | 774/1,000 | 50k | 194k |
| German | ———— | 3mln | 56mln | 106.4mln | 1.12mln | ———— | ———— | ———— | ———— |

Table 1: Left-hand part: Data statistics for Bavarian and Alemannic dialect Wikipedia. Alemannic Wikipedia is bigger than Bavarian, both are magnitudes smaller than standard German Wikipedia. Both Wikis label pages according to fine-grained dialects (# dialects). Center part: the number of sentence pairs manually labelled as similar (labels 4 and 5) out of 1,500 sentence pairs, the number of word pairs manually labelled as correct translations out of 1,000 word pairs. Right-hand part: the overall number of extracted bitext sentences, the overall number of extracted synonyms with a cutoff threshold of 0.8 for **MBERT** alignment probability.

induced bilingual lexicons to gold standard dictionaries (Rapp et al., 2020). Extrinsic evaluation is conducted through cross-lingual downstream tasks (Glavaš et al., 2019). Finally, several factor affect the quality of induced bilingual lexicons: edit distance, contextual and topical similarity between words in source and target languages (Scherrer, 2007; Irvine and Callison-Burch, 2017).

In this project, we apply best practices for *bitext mining* and *bilingual lexicon induction* and demonstrate their strengths and weaknesses in the low-resource settings of *German dialects*.

## 3  Data

Wikipedia offers articles written in more than 300 languages.[2] It is recognized that some parts of Wikipedia are human-translated (Schwenk et al., 2021a); examples are shown in (Table 2). The sentences for our bitext mining and bilingual lexicon induction experiments were extracted from Wikipedia pages in Bavarian[3], Alemannic[4], and German[5]. The Bavarian and Alemannic Wikipedias contain pages marked with nuanced variations in local dialects depending on the region of use. Of the nine dialects of the Bavarian Wikipedia, the most popular is *Westmittelbairisch* (Westmiddlebavarian), with nearly 3k pages. The Alemannic Wikipedia covers 32 dialect varieties, of which *Schwizerdütsch* (Swiss German) is the largest, containing 19k pages. In this work, we do not distinguish between these varieties and treat each Wikipedia as a single corpus.

We used the Wikipedia2corpus[6] tool to extract raw sentences. The texts were split into sentences and tokenized with the SoMaJo sentence splitter and tokenizer[7] (Proisl and Uhrig, 2016). The sentences were filtered by the 5-to-25 token range. Incomplete sentences were removed according to simple heuristics, such as the number of opening and closing brackets or the presence of a bullet point. Sentences containing non-German characters (e.g. letters from Greek, Cyrillic, and Hebrew alphabets) were filtered out. The left-hand part of Table 1 reports the total number of sentences, the number of tokens, and types per language in the resulting Wikipedia datasets. They illustrate the low-resource status of the Germanic dialects compared to the standard. The center part of Table 1 reports the size of manually labelled datasets for both tasks considered: bitext mining and bilingual lexicon induction in Bavarian and Alemannic. The right-hand site of Table 1 reports the sizes of automatically constructed datasets for both tasks in both dialects.

## 4  Bitext mining

**Method.** We start from the assumption parallel sentences are most often found on parallel pages, e.g. pages that are inter-linked between Wikipedias in two languages. We collect interlingual links between pages in dialect Wikipedia and German Wikipedia. Overall, we found 11k parallel pages for Bavarian and 32k parallel pages for Alemannic out of 43k and 71k, correspondingly. Given two parallel pages split into sentences, we embed each sentence with a language model. For each dialect sentence, we retrieve the nearest neighbors using the cosine similarity. Ta-

[2]en.wikipedia.org/wiki/List_of_ Wikipedias, as of 01 Nov 2022
[3]bar.wikipedia.org/, as of 01 Nov 2022
[4]als.wikipedia.org/, as of 01 Nov 2022
[5]de.wikipedia.org/, as of 01 Nov 2022

[6]github.com/GermanT5/wikipedia2corpus
[7]github.com/tsproisl/SoMaJo

ble 2 provides examples of the found parallel sentences and the corresponding cosine similarity values. We aim to select the best-performing embedding model and the optimal cutoff value.

**Models.** We leverage the SentenceTransformer toolkit[8] (Reimers and Gurevych, 2020) for bitext mining. The experiments are with the following mono- and multi-lingual encoders and sentence models released as a part of HuggingFace library[9] (Wolf et al., 2020):

- **MBERT** (Devlin et al., 2019) was pre-trained on Wikipedia data. **MBERT** uses 110k shared across languages WordPiece vocabulary. Note that **MBERT** supports Bavarian and German.
- **GBERT** (Chan et al., 2020) was pre-trained on a range of different German language corpora. Training **GBERT** was carried out with the code-base used to train **MBERT**. Thus **GBERT** uses WordPiece tokenization. The size of vocabulary is 31k. Note that the exposure of **GBERT** to dialects is not mentioned explicitly.
- **GBERT-large-sts-v2**[10] is a version of **GBERT** fine-tuned the semantic textual similarity (STS) datasets of German sentence pairs.
- **LaBSE** (Feng et al., 2022) was pre-trained on the concatenation of mono-lingual Wikipedia and bilingual translation pairs. **LaBSE** uses the WordPiece tokenizer (Sennrich et al., 2016) trained with a cased vocabulary extracted from the model's training set. The vocabulary size is 501,153. **LaBSE** supports German but not its dialects.

We test both `[CLS]` and `[mean]` pooling[11] to obtain sentence representations from **MBERT** and **GBERT**. **GBERT-large-sts-v2** and **LaBSE** are sentence models and can be used out of the box to compute the similarity between sentences. **LaBSE** is the current state-of-the-art-model for bitext mining (Reimers and Gurevych, 2020).

**Human evaluation.** We sampled two random sets of 1,500 bitext instances with **LaBSE** similarity values in the $[0.4; 0.95]$ range to be manually labeled for semantic similarity and further justifications (see next and Appendix for details). We start from the LaBSE model since it is the current state-of-the-art model that has been shown to produce high-quality sentence embeddings (Ham and Kim, 2021). These embeddings capture both semantic and syntactic information, making them useful for a range of natural language processing tasks, including bitext mining. Furthemore, LaBSE was trained on a large-scale multilingual corpus, which makes it more robust and better able to handle variations in language and text structure (Feng et al., 2022).

The annotation schema utilized in our study is a five-point Likert scale, with a score of 5 indicating equivalence between the dialect sentence and the German sentence, and a score of 1 indicating no relation. Annotators were instructed to provide justifications for assigning scores that deviated from 5, by assessing the factual similarity between two given sentences, considering whether one sentence provided more information than the other. Additionally, annotators were asked to identify any significant differences in grammatical structure between the two sentences. The annotation instructions are provided in Section A. The Likert scale is a standardized approach to measuring sentence similarity, providing a more balanced set of response options when compared to binary judgments (Agirre et al., 2012). The annotations were carried out by a native German speaker with a linguistic background and significant exposure to dialects.[12] To ensure the quality of annotations, a smaller sample of 200 sentences was labeled by a second annotator, one of the authors, who is fluent in German.

The annotators were instructed to abstain from labeling sentence pairs with a Likert scale if they lacked a full understanding of the content, if the sentence was written in standard German rather than the dialect, or if the sentence contained a mixture of both. The inter-annotator agreement between the two annotators yielded a score of 0.80/0.78 for exact match and Pearson correlation, respectively, for Bavarian, and 0.9/0.6 for Alemannic. Notably, the primary source of confusion between the annotators was in labeling sentence pairs with scores that were in close proximity,

---

[8] https://www.sbert.net
[9] https://huggingface.co
[10] https://hf.co/deepset/gbert-large-sts

[11] The sentence representation obtained through `[CLS]` pooling uses the `[CLS]` token, while the sentence representation obtained through `[mean]` pooling averages token embeddings.

---

[12] The annotator was hired and received fair compensation according to the local employment regulations.

| Bavarian | German | ✎ | COS |
|---|---|---|---|
| Da Geiselbach speist ob da Omersbachmindung oanige Weiher. | Der Geiselbach speist ab der Omersbachmündung einige Weiher. | 5 | 0.94 |

| Alemannic | German | ✎ | COS |
|---|---|---|---|
| Dr Film verzellt d'Geschichte vume Polizistepaar, dem si Idealismus im Lauf vu dr Handlig schwindet. | Der Film erzählt die Geschichte eines Polizistenpaares, deren Idealismus im Laufe der Handlung schwindet. | 5 | 0.92 |

Table 2: Examples of parallel sentences in Bavarian and German (top) and Alemannic and German (bottom). ✎ denotes a human score (see Section 4 for more details on human evaluation), COS stands for the cosine similarity between **LaBSE** embeddings.

specifically (2,3) and (3,4). Notably, there were no instances in which the annotators disagreed and assigned opposite scores of 1 and 5. Additionally, the annotators were instructed to reject incomplete sentences or those not written in dialect, resulting in the rejection of 83 and 162 sentences, respectively. The remaining 1,417 and 1,338 sentence pairs in Bavarian and Alemannic were included for further analysis.

The results from the human annotation show that the distribution of labels is different for the two dialects: 1,254 sentences were labeled as similar (5) or near similar (4) for Bavarian and almost twice as less, 644 – for Alemannic. In the 250 Alemannic sentences marked with the label 3, the annotator pointed out that bitext sentences differ in minor factual details. Sentences in Bavarian differ less from their German counterparts, so that fewer than 100 sentences are marked as having differences in minor factual details. There are 250/350 Bavarian/Alemannic sentences labeled as using different grammatical structures such as active VS passive, imperfect VS perfect. In summary, based on this annotation study, we conclude that the authors of the Bavarian Wikipedia are more inclined towards literal translation, while the authors of the Alemannic Wikipedia rely less on translation.

**Model comparison.** Many of the retrieved sentence pairs have high similarity values. For instance, **LaBSE** assigns the scores of 0.8 or above to 42% and 24% of the dataset for Bavarian and Alemannic, correspondingly. Overall, the distribution of cosine values tends to be skewed to higher values for all embedders. **GBERT-large-sts-v2** shows the least reasonable performance: the average similarity value is 0.98 and the standard deviation is close to 0.01 for both dialects, leaving no discriminative power to se-

lect a precise cutoff threshold. This may happen due to over-fitting to semantic similarity tasks. The choice of pooling strategy does not affect the performance of **MBERT**: **MBERT+**`[CLS]` and **MBERT+**`[mean]` output strongly correlated cosine values (0.81 and 0.82 for Bavarian and Alemannic). This is not the case for **GBERT**, for which both `[CLS]` and `[mean]` pooling strategies lead to less correlated results ($\approx$0.5 for both dialects). We include the MBERT, GBERT, and LaBSE models in our comprehensive comparison of their abilities in bitext mining and bilingual lexicon induction (Section 5).

The resulting annotated dataset helps to evaluate whether the embedders can judge semantic similarity and assign lower scores to unrelated sentences. At the same time, we may use it to calibrate the cutoff threshold, which distinguishes between similar and unrelated sentence pairs. Figure 3 and Figure 4 in Appendix C show the cosine similarity values, derived with **MBERT+**`[CLS]`, **GBERT+**`[CLS]`, and **LaBSE** models, grouped according to Likert scale values. Although none of these models can divide the data into five groups, there is more evidence that **LaBSE** better separates unrelated sentences (scores 1, 2) from nearly similar or similar sentences (scores 4, 5) leaving somewhat similar sentences (score 3) in between. After careful consideration, we have chosen to use **LaBSE** in subsequent BLI experiments, setting the cutoff for the cosine similarity of nearly similar sentences to 0.7.

**Results.** Our bitext mining efforts resulted in 17k and 50k parallel Bavarian-German and Alemannic-German sentence pairs, respectively, sourced from Wikipedia. These pairs comprise 13.5% and 10% of the total number of sentences in their Wikipedia dumps, as shown in Table 1. After comparing various models, we have determined

that **MBERT** and **LaBSE** are the most closely aligned with human evaluation. This is likely due to **MBERT**'s previous exposure to dialect data, and **LaBSE**'s use of a sentence similarity objective during pre-training.

# 5 Bilingual lexicon induction

**Method.** We use the state-of-the-art awesome-align toolkit[13] (Dou and Neubig, 2021) with **MBERT** and **GBERT** as backbone models. Awesome-align supports an unsupervised mode, so there is no need to fine-tune the models on the parallel data. The word alignments are extracted from parallel sentences by evaluating the similarity between word representations. Awesome-align produces one-to-one alignment by default. When the source dialect sentence uses the perfect tense and the target German sentence uses the preterite tense, in the vast majority of cases, the auxiliary verbs align with the preterite verb.

We feed the extracted parallel dialect-German sentences to the aligner. The outputs are word pairs, in which one of the words is written in dialect and the other in German (see Table 3 for examples of word-level aligned parallel sentences). Each word pair is assigned with alignment probability (see Table 4 for sample output).

Next, we use several strategies to evaluate collected word pairs. We excluded word pairs that contained a non-word token, such as a number, typographical symbol, or punctuation mark. Previous research has demonstrated that the performance of BLI methods is highly dependent on word frequency, with higher frequency source words generally resulting in more accurate translations (Søgaard et al., 2018). To account for this, and to increase coverage of low-frequency words, we employed a stratified sampling approach for word selection in our evaluation. Specifically, we computed the frequency of each dialect word in Wikipedia and divided word pairs into four groups based on quartiles of dialect word frequency. From each group, we randomly selected 250 word pairs for further analysis.

**Dictionary-based evaluation.** To the best of our knowledge, there are no high-quality Bavarian-German or Alemannic-German lexicons, that can be easily accessed for computational experiments, so we turn to community-based resources.

Glosbe[14] is a collection of community-maintained dictionaries, including Bavarian-German and Alemannic-German dictionaries. Since Glosbe does not provide an API, we manually look up German words and record the suggested translations into dialects.

Table 5 shows that the Glosbe dictionary provides better coverage for high-frequency words. The ratio of obtained translation sinks from 29% to 5% from high-frequency words to low-frequency words for Bavarian and from 26% to 4% for Alemannic. The low coverage of the Glosbe dictionary can be partially attributed to the absence of compounds, which are naturally present in Wikipedia writing. For instance, words such as *Laubwoidgebiet* (Bavarian, deciduous forest region) do not exist in Glosbe.

The mismatch between the induced translation and the dictionary-based translation is mainly caused by orthographic variations (see Table 6 for examples, in which both the induced and the Glosbe translations appear to be correct, but different from each other). This is especially evident in Alemannic, where only the 43% of high-frequency word pairs match to induced dictionaries.

**Human evaluation.** In addition to the dictionary-based evaluation, we also performed a human evaluation of the same word pairs using a binary scale to assess semantic similarity. Our aim was to determine whether a German word is a correct translation of a dialect word. The word pairs were presented without the surrounding context, and annotators were given the option to reject a word pair if they did not understand the dialect word. The use of a binary scale was chosen because it simplifies the assessment of semantic similarity and provides a clear indication of whether a word pair is a correct translation or not. The same annotators who participated in the evaluation of the bitext (Section 4) were recruited for the task. We provided annotators with guidelines that are detailed in Appendix B. To assess the level of agreement between annotators, we included a control sample consisting of 200 word pairs for each dialect. The exact match between annotators was high, with a score of 0.96 for Bavarian and 0.85 for Alemannic. Disagreements between annotators were mainly caused by judgments of overlapping words (*Turm – Kirchturm*, steeple – church steeple, in Bavarian).

---

[13]https://github.com/neulab/awesome-align

[14]https://glosbe.com/

| Bavarian to German word alignment |
|---|
| Des Kloster Gunzenhausen is a obgongans Benediktinakloster im Bistum Eichstätt . |
| Das Kloster Gunzenhausen ist ein abgegangenes Benediktinerkloster im Bistum Eichstätt . |

| Alemannic to German word alignment |
|---|
| Um 1267 isch dr Heinrich I . Münch , dr Vater vom Hartung Münch , as Basler Bürgermäister erwähnt worde . |
| Um 1267 wurde __ Heinrich I . Münch , __ Vater von Hartung Münch , als Basler Bürgermeister erwähnt _____ . |

Table 3: Examples of word level alignment in parallel sentences in Bavarian (top) / Alemannic (bottom) and German . Underscore __ stands for unaligned words. **MBERT** is the backbone model.

| Bavarian | German | # | $P$ |
|---|---|---|---|
| Eihgmoant | Eingemeindet | 112 | 0.99 |
| Sidlichn | Südlichen | 71 | 0.96 |
| Augschburg | Augsburg | 39 | 0.91 |
| **Alemannic** | **German** | **#** | $P$ |
| Dytsche | Deutsche | 290 | 0.77 |
| Yywohner | Bewohner | 189 | 0.83 |
| Uniwersidäät | Universität | 126 | 0.95 |

Table 4: Examples of aligned word pairs in Bavarian (top) / Alemannic (bottom) and German. #: the frequency of the word pair. $P$ stands for alignment probability. **MBERT** is the backbone model.

| Q. | Dictionary ✍ | Dictionary ✔ | Human ✎ |
|---|---|---|---|
| **Bavarian**: overall 860 words | | | |
| 1 | 5% | 50% | 85% |
| 2 | 6% | 50% | 95% |
| 3 | 16% | 65% | 90% |
| 4 | 29% | 60% | 65% |
| **Alemannic**: overall 774 words | | | |
| 1 | 4% | 70% | 94% |
| 2 | 5% | 63% | 95% |
| 3 | 9% | 81% | 80% |
| 4 | 26% | 43% | 40% |

Table 5: Dictionary-based evaluation of induced bilingual lexicons, created from sentences, aligned with **LaBSE** and **MBERT** used as the aligner's backbone model. The results are grouped by the frequency quartile of German words, with 1 representing the low-frequency bin and 4 representing the high-frequency bin. Each bin contains 250 words. The percentage of words found in the Glosbe dictionary is denoted by ✍, while ✔ represents the percentage of matched word pairs between the dictionary and induced lexicons. The percentage of word pairs labeled as correct in human evaluation is denoted by ✎.

The evaluation of BLI through human annotation is presented in Table 5. The results indicate that the alignment quality of low-frequency and mid-frequency words is high, with a range of 85% to 95% in both dialects. However, for high-frequency words, the alignment quality drops significantly to 65% in Bavarian and 40% in Alemannic. This decline can be attributed to a higher prevalence of high-frequency prepositions, pronouns, and forms of auxiliary verbs that are often misaligned. Additionally, high-frequency words may contain multiple different spellings of the same word, leading to further noise in the alignment. This effect is more pronounced in Alemannic, where the number of fine-grained dialects is higher compared to Bavarian (as evidenced by Table 1 and the examples in Figure 1).

Interestingly, for mid-frequency words, one of the main sources of errors is the alignment of words that may be used in similar contexts but are not synonyms. For example, the word pair "Soizsään – Mineralquellen" in Bavarian, which translates to "salt lakes - mineral springs" in German, was found to be misaligned. Overall, the annotation study identified 860 and 774 out of 1,000 synonym word pairs between Bavarian and German, and Alemannic and German, respectively.

**Baseline.** We use supervised **MUSE** embeddings (Lample et al., 2018) as a baseline for bilingual lexicon induction. We employ fasttext embeddings, pre-trained on mono-lingual Wikipedia (Bojanowski et al., 2017), with identical character words as seeds. Following the project's guidelines,[15] we set up the dialect embeddings as the

---
[15]https://github.com/facebookresearch/MUSE

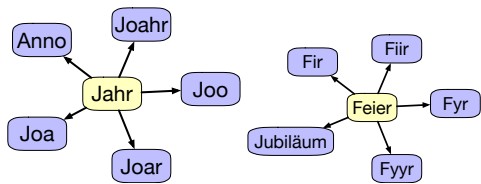

Figure 1: Manually picked examples of one-to-many correspondence from Bavarian-German (left) and Alemannic-German (right) bilingual lexicons. German words are in yellow, dialect words are in blue.

| **Bavarian** | **German** | **Glosbe** $_{(de \rightarrow bar)}$ |
|---|---|---|
| Obapfäjza Zamm | Oberpfälzer Zusammen | Obapfejza Z'samm, zaum |
| **Bavarian** | **German** | **MUSE**$_{(de \rightarrow bar)}$ |
| Vagressade Freizeidzentrum | Vergrößerte Freizeitzentrum | Großhadern Sportpark |
| **Alemannic** | **German** | **Glosbe**$_{(de \rightarrow als)}$ |
| Barlemäntarischi Nobelprys | Parlamentarische Nobelpreis | Parlamentarischi Nobelpreis |
| **Alemannic** | **German** | **MUSE**$_{(de \rightarrow als)}$ |
| Flüssige Epos | Flüssiger Heldengedicht | Wassermolekül Heldenepos |

Table 6: Differences between word pairs induced with **MBERT** and the Glosbe dictionary, **MUSE** synonyms.

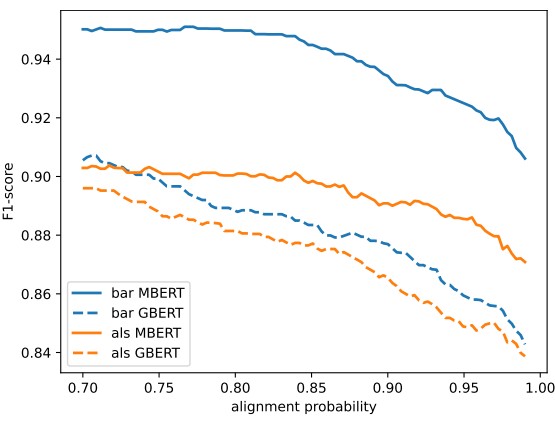

Figure 2: Comparison of two backbone models for Bavarian (blue) and Alemannic (orange). X axis: the cut-off threshold for alignment probability. Y axis: $F_1$ scores. The solid line stands for **MBERT**, and the dashed line stands for **GBERT**. **MBERT** consistently outperforms **GBERT** for both dialects.

source space and German embeddings as the target space. For each dialect word, we retrieve the nearest neighbor according to cosine similarity.

The **MUSE** embeddings retrieve 48 (out of 860) and 74 (out of 774) word pairs (Bavarian/Alemannic), identified as correct translations in the annotation study. Table 6 shows examples of cases, in which **MUSE** embeddings induce words that are different from those induced from bitext. These words have a similar spelling or can be used in similar contexts, but are not synonyms of source dialect words. Note, that the two-step approach for bitext mining and bilingual lexicon induction and the baseline **MUSE** embeddings leverage upon the same data source, namely, Wikipedia. However, our two-step approach leads to inducing more literal synonyms due to accessing larger contexts.

**Model comparison.** We conducted a comparison of two backbone models for the awesome-align toolkit in the binary classification setup. Specifically, we varied the threshold on alignment probability within the range of $[0.7; 0.99]$ and classified word pairs according to whether their probability was above or below the threshold. Negative and positive labels were assigned accordingly. We then compared these predictions to human yes/no labels and computed the $F_1$ score. The results are depicted in Figure 2.

Based on our analysis, it appears that the performance of **MBERT** reaches a plateau within the threshold range of $[0.7; 0.8]$ and gradually decreases as the threshold increases beyond this range. As a result, setting the cut-off threshold at 0.8 represents a reasonable choice. Furthermore, our results suggest that **MBERT** consistently outperforms **GBERT**. The superior performance of **MBERT** may be attributed to several factors, such as the inclusion of dialect data in its pre-training or the larger size of its tokenizer vocabulary.

**Edit distance.** Following prior works (Hangya et al., 2018), we explore the contribution of the edit distance to the word alignment probability. We compute the edit distance and normalize it with the sum of the number of characters in two words divided by two. The correlation coefficient

between the normalized edit distance and the average alignment probability makes $-0.4$ / $-0.38$ and $-0.49/-0.56$ for Bavarian with **MBERT / GBERT** backbones and Alemannic, respectively. This means, first, that the words, spelled similarly have higher chances to be aligned. Second, both backbone models significantly rely on the surface-level similarity between words. In our evaluation, edit distance was utilized solely for the purpose of assessment and not as a baseline. Despite its widespread use, edit distance is computationally expensive and is limited in its ability to capture semantic similarities. In lieu of this, we conducted a comparative analysis with MUSE embeddings, which take into account both surface and semantic similarity to provide a more comprehensive evaluation of the performance of our pipeline.

**Results.** After applying a cutoff threshold of 0.8 for **MBERT** alignment probability, we obtained bilingual lexicons containing 15,000 and 68,000 word pairs for Bavarian and Alemannic, respectively, as summarized in Table 1.

However, the resulting lexicons suffer from a high degree of word form repetition, as multiple dialect spellings are often linked to a single German word (see 1 for an illustrative example). Unfortunately, we were unable to merge different forms of the same word due to the lack of dialect stemmers, lemmatizers, or phonemizers. Words of different parts of speech were sometimes aligned, and we were unable to control for part of speech consistency due to the absence of dialect taggers. Clustering similar word forms presents an interesting avenue for future research.

## 6 Conclusion and Future Work

The project developed a two-stage pipeline for inducing bilingual lexicons for German and its dialects, Bavarian and Alemannic. The first stage involved extracting parallel sentences from public data, specifically Wikipedia, while the second stage used an alignment tool to induce word pairs from these parallel sentences. Both stages relied heavily on pre-trained LLMs, which were calibrated based on the results of annotation studies that judged the semantic similarity between extracted sentences and induced word pairs.

Returning to the research question raised, we may conclude that existing LLMs have a certain capacity for inducing bilingual lexicons. Our results have identified two key factors that in-

fluence their performance: (i) whether the pre-training included multilingual or dialect data, and (ii) whether the model was trained with a task-specific objective. Our evaluations demonstrate that the German GBERT is surpassed in both tasks, indicating that its monolingual pre-training is insufficient to effectively process related dialects. However, the main conundrum that remains is developing linguistic pipelines to process diverse and non-standardized dialect data. The development of dialect-specific tools such as lemmatizers, taggers and phonemizers can help improve the accuracy and consistency of bilingual lexicon induction.

Future work includes exploring the effect of fine-tuning cross-lingual LLMs on German and dialect data for bilingual lexicon induction, differentiating between several Bavarian/Alemannic dialects, and extending the experiments to other German dialects.

**Limitations.** While our study provides a comprehensive evaluation of induced bilingual lexicons for the Bavarian-German and Alemannic-German language pairs, there are some limitations to our approach. These limitations come with the low-resource setup.

**Single domain.** There is no large-scale dialect data source available, so we stick to Wikipedia as almost the only reasonable domain.

**No extrinsic evaluation.** One limitation is the lack of extrinsic evaluation due to the absence of annotated downstream datasets for these language pairs. We relied solely on intrinsic evaluation methods, which limits our ability to assess the usefulness of the induced lexicons in practical settings.

**No multi-word expressions.** Our evaluation focused on the alignment of individual words rather than multi-word expressions (MWEs).

Overall, the two-step pipeline of bitext mining and word aligning has its own disadvantages, such as resulting in one-to-one sentence / word alignment and over-relying on surface-level features.

## Acknowledgements

Thanks to Anna Barwig for her contribution to the project on early stages. Special thanks to members of the MaiNLP lab for their feedback on this paper. This research is supported by ERC Consolidator Grant DIALECT 101043235.

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

## A  Bitext Annotation. Are these two sentences similar?

**Task.** Compare two sentences. One sentence is written in a Bavarian dialect. Another sentence is written in the standard German language. Your task is to compare these two sentences and decide how similar or different they are. You will be asked questions about sentence meaning, if one sentence provides more information than the other, and if the dialect sentence can potentially be a translated version of the German sentence.

**Meaning.** On a scale from 1 to 5, rate how close the meaning of sentences is. Choose the "idk" option if you do not understand the sentence to judge the similarity and skip the rest of the questions. Choose "n/a" if the first sentence is not in a Bavarian dialect and skip the rest of the questions. Choose "incomplete" if the dialect sentence is not complete or some parts of the sentence are missing and skip the rest of the questions. The scores can be interpreted in the following way.

| Label | Explanation |
|---|---|
| idk | I do not understand the dialect sentence. |
| n/a | The dialect sentence is not written in the dialect. |
| incomplete | The dialect sentence is not complete (see below). |
| 1 | The sentences are completely unrelated. |
| 2 | The sentences have minor details in common (shared generic topic). |
| 3 | The sentences refer to same entities, but there are major differences (shared specific topic). |
| 4 | The sentences refer to same entities, but there are minor differences. |
| 5 | The sentences have identical meaning. |

Table 7: The markup schema for bitext annotation.

Try to judge the differences between the sentences from the context. Do you learn the same things from these sentences or not? If one sentence adds more information, is it something really important?
**Incomplete sentences.** might look like these in Table 8 and should be labelled with 5.

| Sentence | Is it complete? |
|---|---|
| Bédouès, Cocurès, Florac, Fraissinet-de-Lozère, La Salle-Prunet, Le Pont-de-Montvert, Saint-Andéol-de-Clerguemort, Saint-Frézal-de-Ventalon, Saint-Julien-d'Arpaon. | **No.** Reason: This looks like a part of a list. |
| House" Haus und des "Mordecai Lincoln House" Haus san historische Gebaide in Springfield und im National Register of Historic Places aufgfiaht. | **No.** Reason: It looks like a few words in the beginning of the sentence are missing. |

Table 8: Examples of incomplete sentences.

**Identical meaning.** We consider sentences like these in Table 2 to have identical meaning.

| Bavarian | German | Label |
|---|---|---|
| Da Geiselbach speist ob da Omersbachmindung oanige Weiher. | Der Geiselbach speist ab der Omersbachmündung einige Weiher. | 5 |
| Am 31. Dezemba 1990 werd Schladerlmühle ois unbewohnt und in Treffelstein aufgonga bezeichnt. | Am 31. Dezember 1990 wird Schladerlmühle als unbewohnt und in Treffelstein aufgegangen bezeichnet. | 5 |
| As Gebiet vo da Metropolitanstod Neapel is a bliabds Reisezui vo in- und ausländischn Touristn. | Das Gebiet der Metropolitanstadt Neapel ist ein beliebtes Reiseziel in- und ausländischer Touristen. | 5 |

Table 9: Examples of sentences with identical meaning

**Factual similarity**. If the sentences do not have identical meaning, choose from the drop down list one of the following explanations, how they differ:
- The dialect sentence misses details.
- The dialect sentence adds details.

| Bavarian | German | Meaning | Factual similarity |
|---|---|---|---|
| Seitm 1. Mai 2008 isa Easchta Buagamoasta vo da Gmoa Hafenlohr. | Seit dem 1. Mai 2008 ist er Erster Bürgermeister der Gemeinde Hafenlohr und Kreisrat im Landkreis Main-Spessart. | 4 | The dialect sentence misses details. Reason: The standard German sentence provides additional information (Kreisrat im Landkreis Main-Spessart). |
| De Gmoa eastreckt se iwa uma 54km². | Die Gemeinde erstreckt sich über etwa 55km². | 4 | Minor factual inconsistency. Reason: 54km² does not equal to 55km², but the numbers are almost the same. |
| Hafenlohr is duach des Zwoate Gmoaedikt am 17. Mai 1818 a Tei vo da Gmoa Hafenlohr gewoadn. | Hafenlohr ist der Hauptort der Gemeinde Hafenlohr. | 3 | Major factual inconsistency. Reason 1: The dialect sentence provides additional information (duach des Zwoate Gmoaedikt am 17. Mai 1818). Reason 2: The dialect sentence uses "Tei". The standard German sentence uses "Hauptort". |
| As Spuin hod a recht wichtige Funktion in da Entwicklung vo Menschnkinda. | Von besonderer Bedeutung ist hier der Verlauf der individuellen Wachstumskurve. | 2 | Minor common details. Factual similarity is not applicable. |
| A Klassika vo humoristischa Litratua is aa da Ignát Herrmann. | Sein Werk ist ein lyrisches Poem, das hochromantisch und hochdramatisch ist. | 1 | Unrelated sentences Factual similarity is not applicable. |

Table 10: Examples of sentence with scores from 1 to 4.

- Different details: both sentences add some new details and miss different details.
- Minor factual inconsistency.
- Major factual inconsistency.
- n/a: if the sentences are completely unrelated, you can not make a judgment about their factual consistency.

Table 10 shows instructions on how to score less similar sentences.

**Grammar differs?** This is a checkbox: mark yes, if there is a difference between the grammar structure of two sentences. If there is no difference or you can not tell it, skip this section.

**Free form comment.** Is there anything else you would like to notice about these two sentences? Can you explain the reason behind your judgment?

## B Bilingual Lexicon Annotation. Is the translation acceptable?

**Task.** This project aims to evaluate bilingual word pairs. Each word pair consists of:

a. a word in Bavarian;

b. a word in Standard German.

The task is to label each word pair as an acceptable translation from Standard German to Bavarian. Label each word pair with:

1. **yes**, if the translation is acceptable;

2. **no**, if it is not acceptable;

3. **idk**, if you can not tell;

4. **check in the box "different part of speech"**, if the two words belong to different parts-of-speech only if you are sure you can tell it without full context.

5. **check in the box "partial match"**, if the words partially match and one word is a part of another (e.g. *Turm - Kirchturm, Sässl – Bürosessel).*

**Free form comment.** Put down free-form comments when necessary.

## C   Comparison of models for measuring sentence similarity

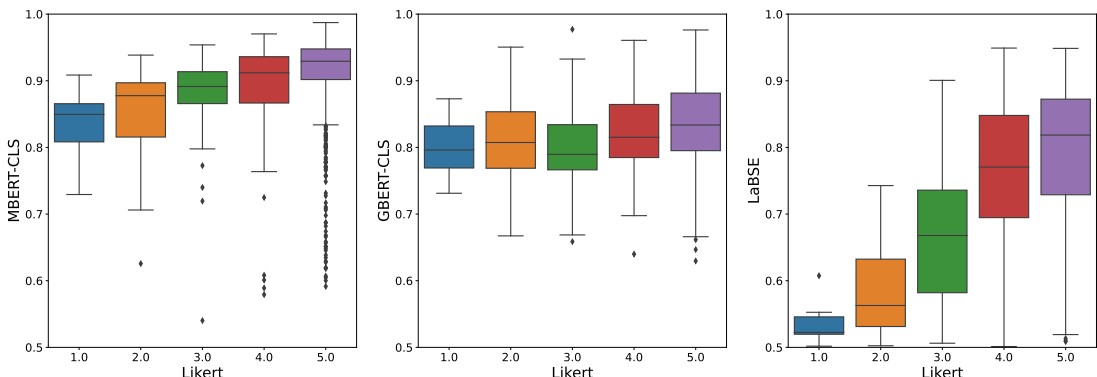

Figure 3: X axis: human scores in sentence similarity for Bavarian. Y axis: Cosine similarity values. The overall number of annotated sentences is 1,417.
Left: **MBERT** + `[CLS]`, middle: **GBERT** + `[CLS]`, right: **LaBSE**. The gap between unrelated and similar sentences is the most evident using **LaBSE** model.

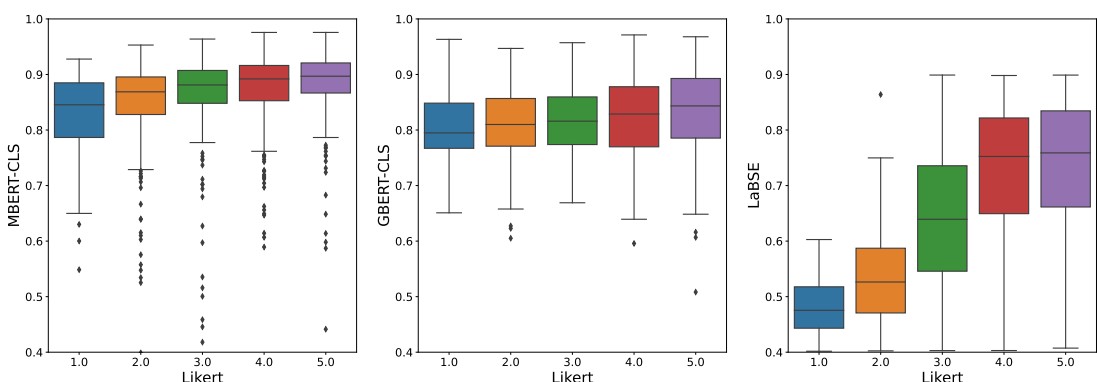

Figure 4: X axis: human scores in sentence similarity for Alemannic. Y axis: Cosine similarity values. The overall number of annotated sentences is 1,338.
Left: **MBERT** + `[CLS]`, middle: **GBERT** + `[CLS]`, right: **LaBSE**. The gap between unrelated and similar sentences is the most evident using **LaBSE** model.