# OpenReview forum: "Low-resource Bilingual Dialect Lexicon Induction with Large Language Models"
_NoDaLiDa/2023/Conference — NoDaLiDa 2023_

### Official Review · Reviewer_CLSY · 2023-02-24
**Induction of German dialect lexicons (Bavarian/German and Alemannic/German) by using bitext and word alignment based on large language models**

**Rating:** 8
**Confidence:** 4

**Review:**

This is a well written paper that is mostly easy to follow. Multiple models are tested. The evaluation is performed both automatically and by human experts. There is thorough discussion and an extensive list of references. The appendix contains the evaluation guidelines for the linguists, which is a good addition. The authors make a good job at explaining the motivations, the task and the conclusions.

My comments mostly relate to details here and there:

* In Section 2, what is [CLS]-pooling? I understand what [CLS] stands for and what pooling is, but are these two really combined here?

* In Section 4, how come LaBSE can be better than models that incorporate dialect data in the training? Is this related to the tokenization performed (BPE?, SentencePiece?), such that similar words can be matched? Or are character n-grams used as in fastText? Or how else can the dialectal OOV words be modeled?

* In the captions of Figures 1 and 2, the labels OX and OY are used. That was new to me, and I found it a bit confusing.

* Table 5 is a bit unclear: If MBERT is compared with MUSE, how should this be read? For the MBERT entries, the right-most column is Glosbe (= gold standard). For the MUSE entries, the right-most column is MUSE (= the method being assessed). This appears to be inconsistent.

* In Section 5, the edit distance method for word alignment is mentioned in passing. How would this method have worked just by itself, instead of MBERT or GBERT? The authors seem to mention edit distance only in order to analyze properties of MBERT and GBERT?

* Additionally, please proofread. There are minor mistakes and typos in many places.





**Paper Type:**

Long paper

---

### Official Review · Reviewer_7U4J · 2023-03-09
**bilingual lexicon experiments**

**Rating:** 3
**Confidence:** 4

**Review:**

In this paper, the authors propose to create bilingual lexicons for German-Bavarian and German-Allemanic combinations. They first extract likely bitext from Wikipedia and then use a word alignment model to create bilingual lexicons. Finally, they evaluate these with human evaluation: for the aligned bitext, they annotate extracted sentence pairs on a 5-point Likert scale, while for the lexicons themselves they annotate whether an extracted pair is correct or not. Finally, they compare the created lexicons with a community constructed bilingual dictionary (Glosbe) and with an embedding-based lexicon induction approach (MUSE).

Much of this paper is dedicated to comparing the sentence similarity models (mBERT, gBERT, and LaBSE). However, it is not automatically clear why this comparison is necessary or how it deals with the actual problem of creating the bilingual lexicon.

However, the main problem is the evaluation of these lexicons. The authors first construct the extracted lexicon and then analyze this data post hoc. Unfortunately, this means that comparing to Glosbe and MUSE is no longer meaningful (see comments). Although Glavaš et al. (2019) are mentioned in the related work, the evaluation strategy does not follow these best practices, i.e., there is no extrinsic evaluation conducted.

I think this paper is currently suffering from a lack of clear research questions to base the experiments on. Other than creating the lexicons themselves, what were the research questions that drive this paper?


Questions/comments:
------
- For the bitext evaluation dataset, you mention you select 2 random samples of 1500 sentence pairs + 200 doubly annotated, but having rejected 83/162 sentences. I am now lost and do not know how many sentences there are for what language and what distribution they have... Figure 1 presumably shows the predicted distributions for the models, but not the true annotated distribution. All of this information should be made completely clear.
- More importantly, I do not understand why we need to evaluate these models on a 5-point Likert scale? Does this granularity give some advantage to creating a bilingual lexicon? I can only really see the benefit of taking highly similar bitext, as anything else would presumably introduce noise...
- Similarly, this makes me question the discussion of skewed cosine values. Why is this a problem/important for the task of bilingual lexicon induction?
- Why is it important to divide the lexicons into frequency bins? What does this tell us?
- We need more information regarding the extracted test set. It seems there are 1000 pairs for each language combination (4 freq bins * 250 pairs), but this should be made explicit and included in a table.
- When comparing the extracted lexicons with Glosbe, what does the column for the percentage of correct word pairs mean? Are the translation pairs in Glosbe incorrect, or do they not match the ones in the extracted lexicon? If they are incorrect, how was this checked?
- When comparing with MUSE, there seem to be a different subset of data used (i.e., the subset identified as correct in the previous human evaluation). This makes the comparison between the two methods of construction a bilingual lexicon incomparable and means that we can not make any conclusions based on the MUSE experiments.
- In the final results, the authors mention that the lexicons contain excessive repetition, many incorrect pairs, and a large variation in spelling. It is not clear if the authors intend to do anything to clean up these lexicons or if this is effectively the end of the road...


**Paper Type:**

Long paper

---

### Decision · Program_Chairs · 2023-03-17

Accept